# BEYOND SOLVING MATH QUIZ: EVALUATING ABILITIES OF LRMS TO *Ask for Information*

## ABSTRACT

The recent development of Large Reasoning Models (LRMs) has demonstrated remarkable problem-solving abilities in mathematics, as evaluated by existing benchmarks exclusively on well-defined problems. However, such evaluation setup constitutes a critical gap, since a genuine intelligent agent should not only know how to solve problems (being a math quiz solver), but also know to *ask for information* when the problems lack sufficient information, enabling proactivity in responding users' requests. To bridge such a gap, we propose a novel dataset consisting of two types of incomplete problems with diverse contexts. Based on the dataset, our systematical evaluation of LRMs reveals their inability in proactively asking for information. In addition, we uncover the behaviors related to overthinking and hallucination of LRMs, and highlight the potential and challenges of supervised fine-tuning in learning such ability. We hope to provide new insights in developing LRMs with genuine intelligence, rather than just solving problems.

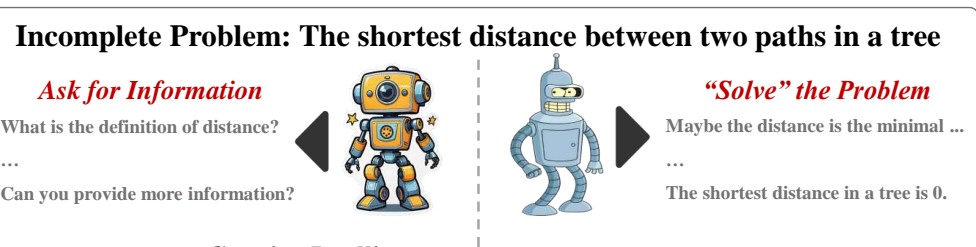

## 1 INTRODUCTION

Mathematical reasoning, a sequence of steps to draw conclusions from given premises (van Eemeren et al., 2001; Angeles, 1981), presents an interesting challenge to artificial intelligence (AI), driving numerous studies and benchmarks (Gao et al., 2025; Lightman et al., 2024; Face, 2025). Recently, Large Reasoning Models (LRMs) (Jaech et al., 2024; Guo et al., 2025) represent remarkable mathematical reasoning abilities in solving competitive problems (Online, 2025), bolstering the conviction regarding the imminent arrival of genuine AI models. However, the creation of genuine AI should not be exclusively evaluated by solving well-defined mathematical problems in existing benchmarks (Lightman et al., 2024; Gao et al., 2025; Face, 2025; Online, 2025). From the standpoint of AI, John McCarthy said that an agent is intelligent "if it can get additional information from the external world when required" (McCarthy & Hayes, 1981). In addition, real-world problems are inherently characterized by incompleteness that fails to provide all necessary information (Zhang et al., 2024; Deng et al., 2023a; Belinda Z. Li, 2025). Such incompleteness makes proactive information-seeking critical for AI assistants to provide truly helpful responses. For example, a user may ask "My living room is 6 meters long. How many tiles (with a side length of 60 centimeter) do I need in total?", and any specific answer based on AI-made assumptions is meaningless as we do not know the width of the user's living room and need ask for this premise.[1] Therefore, being genuine intelligence in mathematics, LRMs should not only solve problems, but also acquire the ability to ***ask for information*** on incomplete problems. Otherwise, an impractical math quiz solver is what we will ultimately develop.

---

[1]Responses of different LRMs are presented in Appendix A.

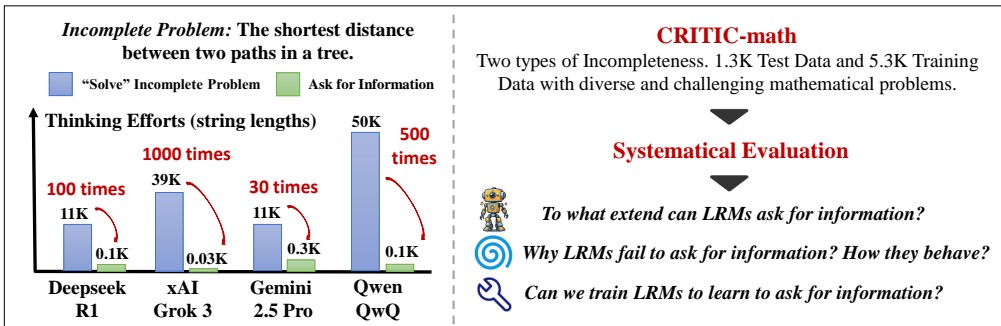

Figure 1: On an incomplete problem (i.e., lacking a definition of "distance"), "solving" the problem exhibits overthinking, manifesting as protracted thinking (measured by thoughts' string lengths), while asking for information reduces thinking efforts. Beyond this example, we provide systematical evaluation of: to what extend can LRMs *ask for information*, why they fail, and how to improve them.

Unfortunately, the ad-hoc example in Figure 1 (left) alerts us current LRMs being far from genuinely intelligent. Confronted with a problem (proposed by Jiayu Yao (2025)) missing a precise definition, LRMs overthink to infer an answer (which is critically impossible because of lacking premises), resulting in significant thinking efforts and delay in responding. This example reveals the limitations of current LRMs in proactively identifying incomplete problems and asking for information. Beyond this example, **to what extend can LRMs ask for information on incomplete problems? Why LRMs fail to ask questions? And can we train LRMs to know how to ask for information**? These questions remain to be systematically evaluated and play vital roles in developing AI models.

Driven by this concern, in this paper, we introduce CRITIC-math (CRItical ThInking of Completeness on mathematical problems, Figure 1 right) to provide systematical evaluation. CRITIC-math contains two types of incomplete problems through rewriting well-defined problems in open-source datasets (Lightman et al., 2024; Gao et al., 2025; Face, 2025). In total, CRITIC-math generates 1.3K test and 5.3K training data, and undergoes manual verification to ensure its quality. Based on CRITIC-math, we reveal the inability of current LRMs in asking for information, and uncover the underlying causes regarding overthinking and hallucination within LRMs' thinking process. We also demonstrate the potential of supervised fine-tuning (SFT) (Ouyang et al., 2022; Muennighoff et al., 2025; Labs, 2025) in learning such ability. In addition, during SFT, we find a dilemma between problem-solving and asking for information, where the current mode of deep-thinking could weaken the ability to ask for information. These results suggest that the existing approach of developing LRMs is biased to only solve math quiz, overlooking the other aspect of intelligence of proactive information-seeking.[2]

In summary, this work makes the following three-fold contributions and offer new insights to develop genuine AI in mathematics that can not only solve problems but also ask for information when needed:

- Propose a new dataset consisting of two types of incompleteness and large-scale problems.
- Based on the dataset, we provide systematical evaluation of state-of-the-art LRMs, uncovering their inability to ask for information and understand how LRMs behave when they fail to ask questions.
- Demonstrate the potential and challenges (a dilemma between the current mode of deep-thinking in solving problems and asking for information) in training LRMs to learn to ask for information.

## 2  CRITIC-MATH

To provide systematical analysis, we introduce CRITIC-math, a new benchmark consisting of two categories of incomplete problems: missing goal and missing premises. We construct CRITIC-math by transforming well-defined problems from open-source datasets into incomplete ones, followed by manual verification t o ensure quality. Table 1 and 2 provide illustrative examples and data statistics. In the following, we adopt Deepseek R1 to synthesize data. To avoid model bias, we use Gemini-2.5 Pro with the same process, and evaluate LRMs performances on data synthesized by both models.

---

[2]We open-source the datasets under Apache 2.0 License using anonymous accounts on HuggingFace. CRITIC-math: `https://huggingface.co/datasets/anonysub25/CRITIC-math`, CRITIC-math-sft: `https://huggingface.co/datasets/anonysub25/CRITIC-math-sft`.

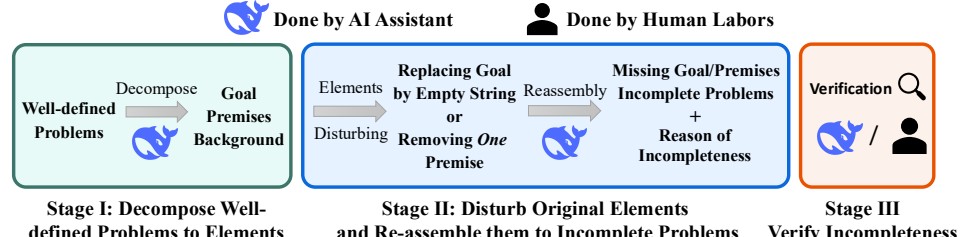

Figure 2: Construction pipeline of CRITIC-math. We rewrite well-defined problems in open-source datasets to incomplete ones in 3 steps: Decomposing, Incomplete Problem Constructing, Verifying.

## 2.1 TYPES OF INCOMPLETE MATHEMATICAL PROBLEMS THAT NEED CLARIFICATION

Reasoning involves a sequence of steps to derive conclusions from premises, and we focus on problems whose goals are assigning a value to a variable (Belinda Z. Li, 2025; Lightman et al., 2024; Gao et al., 2025; Cobbe et al., 2021). Formally, we define reasoning as an assignment function $\mathcal{A}(y|p_1, p_2, ..., p_n)$, where $y$ is the target variable and $p$ are premises. A problem is information-complete if $y$ is well-defined and $\mathcal{A}$ is injective, i.e., only one value can be assigned given the premises. Based on this, as shown in Table 1, we introduce two types of incomplete mathematical problems:

- **Missing Goal**. When the goal is missing, we cannot know the target space of $\mathcal{A}$. Taking the example from Table 1, possible goals can be "how many minutes it takes to cover 1 mile?" or "how many feet can be covered in 1 hour?" Different goals have different values to assign.

- **Missing Premise(s)**. When premises are missing, none or multiple values can be assigned to $y$. Taking the example from Table 1, the bicycle speeds can vary in [1, 10, 20, $\cdots$] feet, different premises can result in different conclusions, and no specific value can be assigned.

Table 1: Two types of incomplete problems and a corresponding example.

| **Well-defined Problem from MATH 500** (Lightman et al., 2024):
A bicycle is traveling at 20 feet per minute. **(Sufficient Premises)**
What is the bicycle's speed expressed in inches per second? **(Clear Goal)** | |
| --- | --- |
| **Type of Incompleteness** | **Incomplete Mathematical Problems** |
| Missing Goal | A bicycle is traveling at 20 feet per minute. |
| Missing Premises | A bicycle is traveling.
What is the bicycle's speed expressed in inches per second? |

## 2.2 CONSTRUCTION OF INCOMPLETE MATHEMATICAL PROBLEMS

**Overview**. CRITIC-math is constructed by rewriting the well-defined problems from open-source datasets. This construction involves three key stages: I) Parsing each problem into its constituent elements (i.e., goal, premises, and Background); II) Disturbing elements and constructing incomplete problems by reassembling these disturbed elements; and III) Verifying the resulting incompleteness. Each stage is elaborated in the following sections, and the overall pipeline is illustrated in Figure 2.

**Data Sources**. For our test set, we adopt problems from test sets of Omni-MATH (Cobbe et al., 2021) and MATH 500 (Lightman et al., 2024). For training set, we use problems from OpenR1-Math (Face, 2025). Given the challenging nature of Omni-MATH, where OpenAI's o1-mini (Jaech et al., 2024) achieves an accuracy of 60% (compared to its 90% accuracy on MATH 500 (Guo et al., 2025)), we limited our selection to Omni-MATH problems with a difficulty level of 4 or lower[3].

**Stage I – Parsing Problems into Goal and Premises**. We prompt Deepseek R1[4] to decompose each problem from the Data Sources into three elements: Goal, Premises, and Background. The Goal represents the result that the problem asks for. Premises are essential information required to achieve the Goal. Background encompasses any remaining information, such as descriptive text and unnecessary examples. All prompts used in the data construction are provided in Appendix B.1.

---

[3]MATH 500 is under MIT License, and Omni-MATH and OpenR1-Math are under Apache 2.0 License.

[4]We select Deepseek R1 for its high performance and suitability for our budget

**Stage II – Constructing Incomplete Problems**. We introduce disturbances to each parsed problem. "Missing Goal" are created by replacing the original goal with empty string. "Missing Premises" are generated by randomly removing *one* entry from original premises list (create up to two variants if the original list contains multiple entries). After disturbances, we prompt DeepSeek R1 to reassemble modified elements and untouched elements into free-form problems, mimics real-world scenarios where elements are not explicitly separated. During this process, Deepseek R1 compares the incomplete problem to its original counterpart and generates the reason for incompleteness.

**Stage III – Human-LLM Quality Verification**. We involve a multi-stage verification process. Specifically, we first prompt DeepSeek R1 to analyze the incomplete problems alongside their reasons for incompleteness (generated in Stage II). If Deepseek R1 judges a problem to be genuinely incomplete, we request it to provide at least two different understanding and possible solutions of the problem. Then, human evaluators provide a final confirmation by reviewing sampled rewritten problems and their AI's judgments. Details on human evaluation and results are in Appendix B.2.

Table 2: Dataset Statistics of CRITIC-math. MP is "Missing Premises", MG is "Missing Goal", and Raw is well-defined problems.

| Source | MP | MG | Raw |
|---|---|---|---|
| CRITIC-math Test | | | |
| MATH 500 | 380 | 234 | 243 |
| Omni-Math | 443 | 254 | 266 |
| CRITIC-math Training | | | |
| OpenR1-math | 3197 | 2153 | 5487 |

**Data Format, Statistics, and Features of CRITIC-math**. Each sample in CRITIC-math is a tuple ($p$, $s$, $r$, $l_d$), where: $p$ represents the incomplete problem; $s$ represents the solution to the original problem where $p$ is rewritten; $r$ denotes the reason of incompleteness; and $l_d$ indicates the difficulty of the original problem ($l_d$ is not applicable in training set, as OpenR1-Math dataset lacks this rating). In addition to the incomplete problems, CRITIC-math also includes the original problems, structured as ($p$, $s$, None, $l_d$), where "None" indicates the absence of incompleteness. Table 2 presents the dataset statistics for CRITIC-math, showing a total of 1311 incomplete problems in the test set and 5350 in the training set[5]. For problems' difficulty, MATH 500 provides five levels (1 to 5), with the corresponding data ratios for "Missing Premises" (MP) being [10.3%, 17.6%, 16.8%, 25.5%, 29.7%] and for "Missing Goal" (MG) being [9.8%, 17.1%, 17.9%, 25.6%, 29.5%]. Omni-Math provides fine-grained levels [1, 1.5, 2, 2.25, 2.5, 3, 3.5, 4], with some levels have little samples. To ensure sufficient samples per level, we organize Omni-Math levels as follows: (1, 1.5) to 1, (2, 2.25) to 2, (2.5) to 3, (3, 3.5) to 4, and (4) to 5. This results in data ratios: MP [24.2%, 16.3%, 10.6%, 10.8%, 38.1%] and MG [25.2%, 15.4%, 11.8%, 11.42%, 36.2%]. In summary, CRITIC-math features two types of incompleteness, more challenging and diverse mathematical problems, a realistic task formulation (i.e., free-form problems), and a training set that allows us to examine how fine-tuning affects the ability of asking for information.

## 3 EVALUATE *Asking for Information* BASED ON CRITIC-MATH

We evaluate LRMs abilities to ask for information on incomplete mathematical problems. Recognizing that Supervised Fine-Tuning (SFT) can adapt LRMs for specific tasks (Muennighoff et al., 2025; Team, 2025a; Xu et al., 2025; Muennighoff et al., 2025), we study two research questions (RQs):

- **RQ1 (Evaluate LRMs)**: To what extend can existing LRMs identify incomplete mathematical problems and raise questions to ask for information?
- **RQ2 (Evaluate SFT Effectiveness)**: Can SFT effectively train LLMs to acquire the ability of *asking for information*? Furthermore, does incorporating the deep thinking from powerful LRMs during SFT enhance LLMs' ability to ask for information?

### 3.1 EVALUATE LRMs (R1)

#### 3.1.1 EVALUATION SETUPS

To conduct a comprehensive evaluation, we introduce two levels of analysis: a coarse level, focused on assessing LRMs' overall performance, and a fine-grained level, designed to understand the thinking characteristics of LRMs and reveal underlying phenomena that why LRMs fail to ask for information.

---

[5]The training problems have been filtered for sampling responses to construct SFT data, cf. Section 3.2.1.

**Prompt Setups**: To examine how LRMs raise clarification questions, we prompt them using two prompts: "Implicit prompt" requires greater proactivity from LRMs in raising questions, and are more closely resemble realistic scenarios where only the problem is provided. In contrast, "explicit prompt" explicitly instructs LRMs to ask for information if needed. The prompts are detailed below.

| Implicit Prompt: | Explicit Prompt: |
|---|---|
| {problem} | # Task to Solve
{query} |
| | # Instruction
If you need ask for information, please raise clarification question and start your response STRICTLY with: "Clarification Question" followed by your questions. |
| | Otherwise, please provide your answer in \\boxed{}. |

**Models**: We use Deepseek-R1 (Guo et al., 2025), Qwen3-plus thinking (Team, 2025b), Claude 3.7 thinking (Antropic, 2025), OpenAI o3-mini (Jaech et al., 2024), and Grok-3-mini (xAI, 2025). These models are the state-of-the-art from various institutions, renowned for advanced reasoning skills.

**Evaluation Metrics**: For the coarse level evaluation of the overall performance regarding raising questions to ask for information and the corresponding thinking efforts, we consider the following metrics, with all evaluations are conducted on both well-defined and incomplete problems.

- Clarification Ratio (CR) is the percentage of responses that raise questions. For "implicit prompts", we utilized LLM-as-a-Judge (Gu et al., 2024) (Deepseek R1) to determine whether a response raised questions. For "explicit prompt", since it requests raising questions by starting with "clarification", we simply check for the presence of the string "clarification".

- Thoughts Lengths in Clarifying (TLC), and Thoughts Lengths in No Clarifying (TLNC) are average thoughts lengths generated by LRMs when raising and not raising questions.

- Clarification Accuracy (ACC) is the overal percentage of responses that raise clarification questions on incomplete problems and not raise clarification questions on well-defined problems.

Fine-grained level evaluates thoughts when fail to ask for information.[6] We split thoughts by '\n\n' Zhang et al. (2025) and use following metrics. ROR and CNR are evaluated by LLM-as-a-Judge[7]. To avoid model bias, we adopt GPT-4.1 and Claude-3.7 as additional judge LLMs (Refer Appendix D.1.2 for results). We observe no significant differences among using different judge LLMs.

- Reflection Step (RS) is the number of steps that contains reflection keywords. Following Hu et al. (2025), we select reflection keywords to include 'alternative', 'wait', 'but', and 'check'.

- Reflection on Incompleteness Ratio (ROR) is the ratio of thoughts whose Reflection Steps specifically aimed at addressing the incompleteness of the problem. In specific, this includes making assumptions about the missing premises and imagining a goal when the goal is missing.

- Clarification Noticing Ratio (CNR) is the ratio of thoughts that identified the need for clarification.

**Implementation Details**. All tested LRMs are accessed via their respective APIs. To ensure deterministic output, we set the temperature to 0 and generate a single response per problem. Appendix D explores experimental results with temperature greater than 0 and multiple responses. Our analysis demonstrates there is no significant differences between sampling one and multiple responses. In addition, Appendix C details the implementation of our LLM-as-a-Judges. Finally, when calculating the thoughts length, we adopt different methods depending on the return format of different API's. For Deepseek R1, we measure the string lengths of the thoughts returned by APIs. For other LRMs, we directly use the number of "reasoning tokens" reported in the API's usage data.

### 3.1.2 RESULTS & DISCUSSION

Table 7 presents the results of our coarse-grained evaluation, while Table 9 details the fine-grained evaluation results[8]. An integrated analysis on both tables yields the following observations:

---

[6]This is only applicable on Deepseek-R1 and Qwen3 as other APIs do not return the thoughts contents.

[7]For longer thoughts, only the first ten steps are evaluated.

[8]In the main page, we report results on dataset synthesized by Deepseek R1. Please refer Appendix D.1.1 for results on dataset synthesized by Gemini-2.5 Pro. We observe no significant differences on these two datasets.

Table 3: Coarse-grained assessment of overall performance. For readability, TLCs/TLNCs are reported as the multiples of the TLNC on well-defined problems, with the absolute lengths shown in parentheses. Multiples greater than 1 are marked in red; otherwise, in green. Results obtained using "implicit prompt" are indicated with gray lines; otherwise, with white.

| Models | Missing Premises | | | Missing Goal | | | Well-Defined | | | ACC |
|---|---|---|---|---|---|---|---|---|---|---|
| | CR | TLC | TLNC | CR | TLC | TLNC | CR | TLC | TLNC | |
| CRITIC-math (from Math 500 Data Source) | | | | | | | | | | |
| Deepseek R1 | 48.68% | 0.61 | 2.11 | 48.72% | 0.49 | 1.05 | 1.23% | 3.08 | 1 (6387) | 62.89% |
| | 21.05% | 2.35 | 2.15 | 4.27% | 1.73 | 1.34 | 0.41% | 4.24 | 1 (7973) | 38.74% |
| Qwen3 Plus | 51.58% | 0.40 | 1.78 | **58.12%** | 0.41 | 0.99 | 0.00% | nan | 1 (3578) | **67.09%** |
| | 21.58% | 1.81 | 1.49 | 35.48% | 1.56 | 1.07 | 0.00% | nan | 1 (4087) | 39.91% |
| o3 mini | 50.70% | 1.00 | 2.07 | 35.04% | 1.34 | 1.09 | 0.41% | 5.90 | 1 (694) | 60.33% |
| | 27.37% | 1.43 | 2.63 | 19.23% | 0.93 | 1.23 | 0.00% | nan | 1 (738) | 45.74% |
| Grok 3 mini | **55.70%** | 0.49 | 1.52 | 48.72% | 0.48 | 0.97 | 0.00% | nan | 1 (1525) | 66.39% |
| | 34.21% | 1.14 | 1.27 | 15.81% | 0.68 | 1.06 | 0.00% | nan | 1 (1587) | 47.84% |
| Claude 3.7 | 53.42% | 0.42 | 2.07 | 44.87% | 0.16 | 0.93 | 2.47% | 0.02 | 1 (4818) | 63.59% |
| | 33.68% | 0.66 | 1.42 | 11.97% | 0.56 | 0.94 | 0.41% | 6.01 | 1 (3882) | 46.44% |
| CRITIC-math (from Omni-Math Data Source) | | | | | | | | | | |
| Deepseek R1 | 44.02% | 0.34 | 1.27 | 53.54% | 0.27 | 0.83 | 3.76% | 0.43 | 1 (14177) | 60.96% |
| | 18.74% | 1.71 | 1.41 | 3.15% | 1.70 | 1.01 | 3.38% | 2.53 | 1 (15583) | 36.14% |
| Qwen3 Plus | 46.73% | 0.30 | 1.34 | **57.87%** | 0.24 | 1.01 | 5.64% | 0.54 | 1 (6284) | 62.82% |
| | 27.65% | 1.47 | 1.15 | 9.45% | 1.98 | 0.95 | 4.14% | 1.54 | 1 (6740) | 37.80% |
| o3 mini | 48.31% | 0.63 | 1.37 | 40.16% | 0.45 | 0.68 | 6.39% | 0.38 | 1 (2085) | 58.67% |
| | 20.32% | 0.86 | 1.65 | 20.87% | 0.33 | 0.88 | 2.63% | 0.50 | 1 (2143) | 41.74% |
| Grok 3 mini | **53.05%** | 0.36 | 1.18 | 55.38% | 0.38 | 0.89 | 8.27% | 0.37 | 1 (2433) | **64.28%** |
| | 31.83% | 1.00 | 1.07 | 20.47% | 0.52 | 0.90 | 3.38% | 1.55 | 1 (2379) | 46.73% |
| Claude 3.7 | 50.34% | 0.31 | 1.89 | 46.46% | 0.11 | 1.10 | 7.89% | 0.02 | 1 (7094) | 60.85% |
| | 23.48% | 1.34 | 1.97 | 14.57% | 0.58 | 1.26 | 2.26% | 2.64 | 1 (4642) | 41.64% |

Table 4: Fine-grained analysis of LRMs *when fail to ask for information*. We report RS on well-defined problems for comparison. Results obtained using "implicit prompt" are indicated in gray; otherwise, in white. We average results of two data sources considering limited pages.

| Model | Missing Premises | | | Missing Goal | | | Well-Defined |
|---|---|---|---|---|---|---|---|
| | RS | ROR | CNR | RS | ROR | CNR | RS |
| Deepseek R1 | 36.58 | 74.04% | 21.22% | 17.21 | 97.68% | 3.78% | 23.39 |
| | 50.38 | 77.27% | 24.33% | 29.55 | 94.77% | 1.06% | 26.41 |
| Qwen3 Plus | 79.79 | 76.43% | 18.57% | 50.37 | 98.45% | 4.88% | 47.79 |
| | 84.51 | 81.40% | 28.35% | 59.28 | 94.60% | 3.80% | 87.39 |

**Overall, LRMs lack the ability to *Ask for Information*.** In Table 7, LRMs achieve significantly low clarification ratios (CRs, around 25%) and accuracies (ACCs, around 40%) when prompted with only problems ("implicit prompt"), and explicitly instructing LRMs to ask for information ("explicit prompt") can improve CRs from 25% to 50% and ACCs from 40% to 65%. These results indicate that LRMs lack the ability to proactively ask for information. In most cases, LRMs act as math quiz solvers to "solve" problems regardless whether the problem is complete or not. Even given "explicit prompt", CRs are around 50%, indicating that LRMs struggle to effectively ask for information. These phenomena are even more pronounced on difficult problems. As detailed in Appendix D.1.3, we observe a negative correlation between problem difficulty and CRs: CRs decrease on harder problems. Unexpectedly, in Appendix D.1.3, we find that when LRMs ask for information, their questions are rather accurate that target at the incompleteness. For such, we hypothesize that LRMs actually lack the ability to ask rather than to notice the incompleteness. LRMs tend to "solve" a problem even know it is incomplete. Our fine-grained analysis further support this hypothesis.

**Fine-grained Evaluation** assesses LRMs' behavior modes when LRMs fail to ask for information. In total, we identify two major failure modes (*thoughts-to-answer unfaithfulness* and *overthinking*) on missing premises problems and one dominate failure mode (*hallucination*) on missing goal problems.

- **Thoughts-to-Answer Unfaithfulness When Missing Premises**. CNR is the ratio of thoughts that have recognized the need to ask. The results in Table 9 suggest that, confronted with missing premises problems, in approximately 20% of failure cases, LRMs recognize the need to ask for information, but ultimately fail to do so. We provide a case for illustration below, where the thought recognizes the necessity ("*the best approach*") and provides a candidate question ("*Are there any specific ...*"). However, the LRM fails to follow its thought, instead generating a answer. This exemplifies a type of thoughts-to-answer unfaithfulness in LRMs, and further support our hypothesis that LRMs know the incompleteness but tend to be math quiz solvers rather than be proactive to ask for information.

> **Thoughts**: If I don't ask, I might give the wrong answer. So the best approach here is to request more details. Clarification Question: Are there any specific conditions given for $\cdots$
>
> ---
>
> **Answers**: To determine $\cdots$ **Final Answer**: If the equation has equal roots, then $\boxed{\pm 2}$.

- **Overthinking when Missing Premises**. As Table 7 shows, confronted with missing premises problems, TLNCs (thoughts lengths in no clarifying) significantly increase. The results in Table 9 further reveal that when LRMs fail to ask for information, their thoughts exhibit higher RSs (reflection steps). These thoughts, as measured by RORs (Reflection on Incompleteness Ratio), contains reflection steps that focus on the incompleteness in roughly 75% cases. In together, if problems lack premises, LRMs tend to address such deficiency through their internal thinking, failing to ask for information, leading to significant delays. In contrast, TLCs (thoughts lengths in clarifying) are much shorter, demonstrating the efficiency of asking questions. In addition, we find the overthinking to be more pronounced on easy problems (detailed in Appendix D), aligning with previous findings (Chen et al., 2024b). Such overthinking also supports our hypothesis, as overthinking on incompleteness indeed indicates that LRMs have noticed the incompleteness. However, LRMs more likely to rely on internal thinking to address the incompleteness and "solve" the problem, rather than asking for information.

- **Hallucinations when Missing Goals**. As shown in Table 9, in more than 90% cases (RORs), the thoughts address the incompleteness by imagining a goal. We consider this behavior as instruction inconsistency (Huang et al., 2025b), where the output deviates from users' directive. Interestingly, TLNCs are shorter compared to well-defined problems, suggesting that LRMs tend to imagine goals that can be quickly addressed. For illustration, refer to Appendix D for case studies.

Unfaithfulness and overthinking explain 80% failures in missing premises, and hallucination explains 95% failures in missing goals. We leave analyzing other minor cases of behaviors to future works. These results help us understand how and why LRMs fail to ask questions: current LRMs are more like math quiz solvers, tending to "solve" incomplete problems by three major kinds of crooked behaviors instead of asking for information even explicitly prompted and recognized the incompleteness.

## 3.2 EVALUATE SFT EFFECTIVENESS (RQ2)

### 3.2.1 EVALUATION SETUPS

**SFT Training Data**. Following the method of distilling LLMs (Hsieh et al., 2023; Hinton et al., 2015), we use Deepseek R1 with "explicit prompt" to generate data for SFT. Specifically, for each problem in the training set of CRITIC-math: (1) For well-defined problems, we store answers and thoughts only if the answers are correct (determined by Math-Verify (Kydlíček)). (2) For incomplete problems, we store answers and thoughts if the answers raise clarification questions (determined by string-matching on "clarification"). The training set was filtered to include only well-defined problems with correct answers and incomplete problems with answers raising clarification questions. This process results in a total of 10.8K SFT samples.

**Implementation Details**: Using Qwen3-8B-Base, we train two models and their variants: **CRITIC-Qwen**, using answers only, and **CRITIC-Qwen-thinking**, using both thoughts and answers. For ablation studies, we train **CRITIC-Qwen$_{W/I}$** and **CRITIC-Qwen-thinking$_{W/I}$** to analyze the impact of problem types, where "W" indicates trained on well-defined problems and "I" indicates trained on incomplete problems. We use OpenRLHF (Hu et al., 2024) to SFT for 1 epoch, with a learning rate of 1e-5 and batch size of 64.[9] Training CRITIC-Qwen took 10 minutes and CRITIC-Qwen-thinking took 60 minutes on 8 NVIDIA A800s. We use vLLM (Kwon et al., 2023) to infer responses. We

---

[9]Other hyperparameters, like the learning rate schedule and warmup steps, are kept at their default values.

conduct additional experiments using Llama-3.1-8B-Instruct. Please refer Appendix D.2 for detail results. We observe no significant differences in the results between the choices of backbone LLMs.

**Evaluation Metrics**: We follow the metrics in Section 3.1.1. Notably, according to the training data-format, we prompt SFT models by "explicit prompt" according to training data-format. To further demonstrate the effectiveness of SFT in improving mathematical reasoning, we introduce Solved Ratio (SR), defined as the percentage of answers that correctly solve the well-defined problems.

### 3.2.2 RESULTS & DISCUSSION

Table 5: Coarse level evaluation of SFT models. *CRITIC-Qwen T* denotes CRITIC-Qwen-thinking. *Qwen3-8B T* denotes the official LRM (also fine-tuned from Qwen3-8B-Base) with thinking enabled.

| Model | Missing Premises | | | Missing Goal | | | Well-Defined | | | | ACC |
|---|---|---|---|---|---|---|---|---|---|---|---|
| | CR | TLC | TLNC | CR | TLC | TLNC | CR | TLC | TLNC | SR | |
| Math 500 Subset | | | | | | | | | | | |
| CRITIC-Qwen | **78.42%** | / | / | **94.87%** | / | / | 4.12% | / | / | 73.39% | **87.86%** |
| CRITIC-Qwen T | 57.37% | 0.22 | 2.21 | 62.82% | 0.25 | 0.93 | 1.23% | 1.59 | 1 (4554) | 80.83% | 70.60% |
| Qwen3-8B T | 51.58% | 0.39 | 1.71 | 50.43% | 0.41 | 1.05 | 0.00% | nan | 1 (3853) | **97.53%** | 64.99% |
| Omni-Math Subset | | | | | | | | | | | |
| CRITIC-Qwen | **77.88%** | / | / | **97.64%** | / | / | 13.16% | / | / | 41.13% | **85.57%** |
| CRITIC-Qwen T | 56.66% | 0.14 | 1.51 | 69.29% | 0.14 | 1.09 | 7.52% | 0.32 | 1 (8895) | 54.88% | 69.89% |
| Qwen3-8B T | 45.82% | 0.29 | 1.32 | 54.72% | 0.30 | 0.90 | **5.26%** | 0.32 | 1 (7027) | **83.74%** | 61.68% |

Table 6: Fine-grained analysis of SFT model *when fails to ask for information*. Only CRITIC-Qwen-thinking can be evaluated. we average results of two data sources considering limited pages.

| Model | Missing Premises | | | Missing Goal | | | Well-Defined |
|---|---|---|---|---|---|---|---|
| | RS | ROR | CNR | RS | ROR | CNR | Rs |
| CRITIC-Qwen T | 82.84 | 74.01% | 21.75% | 44.10 | 92.12% | 6.06% | 41.30 |

Table 13 presents the results of our coarse-grained evaluation, while Table 6 details the fine-grained evaluation results. An integrated analysis on both tables yields the following observations:

**SFT Improves the Ability to *Ask for Information***. CRITIC-Qwen achieves higher CRs on incomplete problems. Although more false positives on well-defined problems, the accuracies are 87.86% and 85.57%, surpassing the highest accuracy of close-source LRMs (67.09% and 64.28% achieved by Grok 3 mini). CRITIC-Qwen-thinking also surpasses close-source LRMs. In addition, both SFT models surpass Qwen3-8B T, which fine-tuned from the same backbone but not tailored for asking for information. These results demonstrate the effectivenss of SFT in learning to ask for information.

**A Dilemma between Deep Thinking and *Asking for Information***. As shown in Table 13 and 6, when SFT with thinking process, solved ratios on well-defined problems increase, demonstrating improved mathematical reasoning. However, CRs on incomplete problems significantly decrease, mirroring the overthinking and hallucinations observed in the closed-source LRMs discussed in Section 3.1.2: large TLNCs/RSs and high RORs indicate that CRITIC-Qwen-thinking overthinks on missing premises. Additionally, the barely changed TLNCs and high RORs on missing goal problems suggest that CRITIC-Qwen-thinking imagines and accomplishes a goal not specified by the input.

These results demonstrate a dilemma between the current mode of deep-thinking, which encourages self-reflections in solving problems (Muennighoff et al., 2025; Guo et al., 2025), and asking for information. Given high RSs and RORs in Table 9 and 6, we argue that one root cause is that the current deep-thinking makes LRMs treat missing information as an internal issue ( poor understanding of input) and stimulating self-reflections. This represents a flaw in current research: the development of LRMs is significantly biased towards problem-solving, which overemphasizes extensive internal thinking to address any issue (including the incompleteness) and "solve" problems. Ablation studies, detailed in Appendix D, further support our analysis, where CRs of CRITIC-Qwen-thinking$_I$ increase as its self-reflections are less strengthened by learning on well-defined problems. In addition, there is no significant differences between CRITIC-Qwen-thinking$_W$ and CRITIC-Qwen-thinking on well-defined problems, indicating that asking for information has limited side-effects on solving problems.

## 4 RELATED WORKS

**The Ability to Identify Incomplete Problem**. In psychology, a child's ability to identify missing information in mathematical problems is considered an important indicator of their intelligence (Dempster & Corkill, 1999; Edens & Potter, 2008), reflecting their use of schematic knowledge (understanding problem structures) (Low & Over, 1989) and metacognition (supervising one's thoughts) (Lai, 2011; Medina et al., 2017). Current LRMs demonstrate advanced reasoning abilities, inspiring recent exploration into their capabilities from a cognitive perspective (Gandhi et al., 2025). Our work examines the behaviors and limitations of LRMs in identifying incomplete mathematical problems, revealing challenges in developing AI with human-level cognition. In addition, proactively identifying and gathering missing information is a fundamental philosophical problem in AI (McCarthy & Hayes, 1981), constituting central topics in various topics (Ren et al., 2021; Gal et al., 2017; Sutton et al., 1998; Curtis et al., 2024; Phiquepal & Toussaint, 2019). Following this tradition, our work suggests that LRMs currently over-rely on internal thinking and struggle to effectively ask for information yet.

**Clarifying User Requests**. Incompleteness frequently appears in natural language due to various factors (Piantadosi et al., 2012; Wasow et al., 2005; Degani & Tokowicz, 2010), and recognizing and resolving it has posed long-standing challenges for developing AI models (Dreyfus, 1972; Anwar et al., 2024; Huang et al., 2025a). In the era of LLMs, extensive research has explored ambiguity in various scenarios, including conversation (Zhang et al., 2024), semantics Kuhn et al. (2023), question-answering (Min et al., 2020), and chatbots Chen et al. (2024a). To improve abilities of LLMs to handle ambiguity, recent approaches utilize Chain-of-Thought, few-shot prompting, or training adapters (Chen et al., 2024c; Deng et al., 2023b; Kuhn et al., 2022; Cole et al., 2023) to develop assistants that proactively understand users' intents. In this paper, given the current trending research of LRMs, we extend the research scope of clarifying questions from general domain to mathematics, and argue that asking for information on incomplete problems stands for a critical aspect to develop genuine intelligence. We also explore the potential and challenges in learning such ability, hoping to providing insights to develop LRMs that can ask for information.

**Benchmarks in Asking for Information**. Existing benchmarks primarily focus on users requests in general domains like task-oriented dialogue (Budzianowski et al., 2018; Rastogi et al., 2020; Zhang et al., 2024; Basile et al., 2021; Wan et al., 2023). Limited efforts are dedicated to analyzing LRMs in handling incomplete math problems. Given the increasing prevalence of LRMs and current exclusive evaluation on well-defined problems, we propose a new dataset to systematically evaluate LRMs in asking for information on incomplete problems. We acknowledge a concurrent work, QuestBench (Belinda Z. Li, 2025) (released March 28, 2025 on Arxiv), that examines LLMs' information-gathering in reasoning tasks. However, QuestBench only evaluates can LLMs pick up the correct question among several candidate choices. Such setting fails to evaluate the proactivity of asking for information, and does not discuss the relationship between problem-solving and asking questions, where a dilemma is found in this work. CRITIC-math aims to provide more comprehensive evaluation of LRMs in proactively asking for information, extending current scope of solving problems.

## 5 CONCLUSIONS

Existing evaluation of LRMs exclusively focuses on solving well-defined mathematical problems, ignoring another critic aspect of genuine intelligence regarding asking for information on incomplete problems. To bridge such gap, we introduce a new benchmark, CRITIC-math, provide systematical evaluation, and find LRMs' inability to identify incompleteness and proactively seek clarification. Our observed dilemma between the current mode of deep thinking and asking for information also demands a fundamental shift in how we develop LRMs. We hope to offer guidance and insights for developing LRMs that transcend solely being math quiz solver, evolving into genuine intelligent agents capable requiting information, navigating uncertainties inherent in real-world scenarios.

Moving forward, we intend to broaden the scope of our research to new domains, such as computer programming. This expansion, however, necessitates overcoming several challenges. Chief among them are the increased contextual diversity—spanning multiple programming languages and frameworks—and the pronounced scarcity of datasets structured suitably for evaluating information-seeking tasks. Such datasets, which are essential for our methodology, are not yet readily available. Nevertheless, we are committed to advancing this line of inquiry as a vital progression of our work.

## 6 ETHICS STATEMENT

This paper focuses on mathematical reasoning and our major argument is that a genuine intelligence should not know solving problems, but also asking for information. We affirm that this study fully complies with the ICLR Code of Ethics. All data are sourced from public, open-source repositories, obviating the use of human participants or sensitive personal data. All procedures, including any human-in-the-loop verification performed under controlled conditions.

## 7 REPRODUCIBILITY STATEMENT

To facilitate reproducibility, we publicly release the full dataset. CRITIC-math: `https://huggingface.co/datasets/anonysub25/CRITIC-math`, CRITIC-math-sft: `https://huggingface.co/datasets/anonysub25/CRITIC-math-sft` (both under Apache 2.0 License). We have also provided the detail experimental setups in the this paper. In addition, the corresponding code are provided in the supplementary files and will be open-sourced.

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

## A    RESPONSES OF DIFFERENT LRMS

In this section, we present the responses of different LRMs in responding the question "My living room is 6 meters long. How many tiles (with a side length of 60 centimeter) do I need in total?" Among LRMs, only o3-mini asks for information. The others make assumptions to solve the problem.

---

**o3-mini:** To figure out the total number of tiles you'll need, I also need the width of your living room (in meters). With both length and width, we can compute the total area and then divide by the area of one tile (0.6 m × 0.6 m). What's the width of the room?

---

**Deepseek R1:** The user's living room is 6 meters long, and the tiles have a side length of 60 centimeters (0.6 meters). To determine how many tiles are needed in total:

1. **Convert the length of the living room to centimeters**:
6 meters = 600 centimeters.
2. **Calculate the number of tiles along the length**:
Each tile is 60 centimeters long, so the number of tiles needed along the length is:

$$\frac{600 \text{ cm}}{60 \text{ cm/tile}} = 10 \text{ tiles}$$

3. **Consider the total number of tiles**:
Since the width of the living room is not provided, the problem is ambiguous. However, if we assume the question is asking how many tiles fit along the 6-meter length, the answer is 10 tiles.

Given the ambiguity but following the straightforward calculation for the length provided, the final answer is:

$$\boxed{10}$$

---

**Gemini 2.5 pro:** To figure out the total number of tiles, we need both the length and the width of your living room. You've given the length as 6 meters.
First, let's make sure our units are the same:
Room length: 6 meters = 600 centimeters
Tile side length: 60 centimeters
Tiles along the length: Number of tiles = Room length / Tile side length Number of tiles = 600 cm / 60 cm = 10 tiles
So, you would need 10 tiles to go along the 6-meter length of your room.
To find the total number of tiles, you'll also need the width of your room.
Let's say, for example, your room is also 6 meters wide (making it a square room):
Room width: 6 meters = 600 centimeters
Tiles along the width: 600 cm / 60 cm = 10 tiles
Total tiles for a 6m x 6m room: 10 tiles (length) * 10 tiles (width) = 100 tiles
If your room has a different width, please provide that dimension, and I can calculate the exact total for you. For instance, if your room was 3 meters (300 cm) wide:
Tiles along the width: 300 cm / 60 cm = 5 tiles
Total tiles: 10 tiles (length) * 5 tiles (width) = 50 tiles
In summary: You need 10 tiles to span the 6-meter length. Please provide the width of your living room to calculate the total number of tiles needed.

**Qwen3:** To determine how many tiles are needed to cover the floor of a living room that is **6 meters long**, we need to calculate the **total area** of the room and divide it by the **area of a single tile**.

—

### Step 1: Convert Units for Consistency
Each tile has a **side length of 60 cm**, which is equivalent to:

$$60 \, \text{cm} = 0.6 \, \text{m}$$

So, the **area of one tile** is:

$$0.6 \, \text{m} \times 0.6 \, \text{m} = 0.36 \, \text{m}^2$$

—

### Step 2: Determine the Area of the Room
The problem only gives the **length** of the room as 6 meters. To compute the **area**, we also need the **width**. However, since the width is not provided, and the problem asks for the **total number of tiles**, it is reasonable to **assume** that the room is **square** (i.e., length = width = 6 meters), which is a common assumption in such problems when only one dimension is given.
So, the **area of the room** is:

$$6 \, \text{m} \times 6 \, \text{m} = 36 \, \text{m}^2$$

—

### Step 3: Calculate the Number of Tiles
Now, divide the total area of the room by the area of one tile:

$$\frac{36 \, \text{m}^2}{0.36 \, \text{m}^2/\text{tile}} = 100 \, \text{tiles}$$

—

### Final Answer

$$\boxed{100}$$

---

**Claude 3.7:** To solve this problem, I need to find how many 60 cm tiles are needed to cover a length of 6 meters.

First, I'll convert the measurements to the same unit:
- Living room length = 6 meters
- Tile side length = 60 centimeters = 0.6 meters

To find the number of tiles needed, I'll divide the room's length by the tile's side length:

Number of tiles = Length of room ÷ Side length of tile
Number of tiles = 6 m ÷ 0.6 m = 10 tiles

Therefore, you need 10 tiles to cover the 6-meter length of your living room.

## B CRITIC-math CONSTRUCTION AND CASE STUDIES

### B.1 PROMPTS UTILIZED IN CONSTRUCTING CRITIC-math

CRITIC-math is constructed by rewriting the well-defined problems from open-source datasets. This construction involves three key stages: I) Parsing each problem into its constituent elements (i.e., goal, premises, and Background); II) Disturbing elements and constructing incomplete problems by reassembling these disturbed elements; and III) Verifying the resulting incompleteness.

This appendix details prompts used for each stage of our process (DeepSeek R1 was used). Specifically: Prompt in Figure 3 is used for parsing problems into goals, premises and background;. Stage II – Constructing Incomplete Problems – is mainly completed automatically and Prompts in Figure 4 and B.1 is used for reassembly and generating reason of incompleteness. Finally, Prompt in Figure 6 is used for verify the incompleteness.

---

Given a TEXT, its "Goal", "Necessary Information", and "Background" are as follows:
- "Goal" refers to the task that the TEXT requires to solve. "Goal" should be short and concise.
- "Necessary Information" and "Background" refer to everything apart from the Goal, including data, facts, examples, etc. Among them:
– "Necessary Information" refers to the contents that are strictly needed to accomplish the Goal.
– "Background" refers to the other contents that can be ignored, such as examples and descriptions.

Based on the above definitions, please SPLIT the following TEXT into "Goal", "Necessary Information", and "Background".

\*\*\*
TEXT:

problem
\*\*\*

If Background is None, leave Background empty.
Split long Necessary Information to short items.
For "Necessary Information", provide a numbered list.

\*\*Requirements\*\*
1. \*\*You should EXACTLY COPY contents from the TEXT to Goal, Necessary Information, and Background!!!\*\*
2. \*\*DO NOT ADD, REWRITE, REPHRASE, ANY CONTENT WHEN COPYING TEXT to YOUR OUTPUTS!!!\*\*
3. \*\*KEEP ALL SYMBOLS, such as EMPTY LINES, SPACES, MATH SYMBOLS, and FORMATTING SYMBOLS WHEN COPYING TEXT to YOUR OUTPUTS!!!\*\*
4. \*\*Goal, Necessary Information, and Background SHOULD NOT OVERLAP!!!\*\*
5. \*\*Goal, Necessary Information, and Background SHOULD COVER ALL contents of the TEXT!!!\*\*

---

Figure 3: The Prompt to Parse Problem into Goal and Premises.

# Goal

{goal}

# Information

{information}

# Background

{background}

# Instruction

DO NOT consider rationality, assemble the Goal, Information, and Background into a coherent text to get an Unclear Question.
Because the Unclear Question contains Missing Information compared with the Original Question shown below, the Unclear Question should not be able to be answered directly.

***
Original Question:

{original_question}
***

Provide the reasons that why the Unclear Question is Unclear (i.e, can not be answered directly).
Generate your output STRICTLY in the following format.
Unclear Question:
Reason of Unclearness:

# Requirements

1. **The Unclear Question SHOULD NOT CONTAIN the title of Goal, Information, and Background.**
2. **The Unclear Question SHOULD CONTAIN ALL contents in the Goal, Information, and Background (if the Information and Background are not empty).**
3. **DO NOT ADD ANY content that is not included in the Goal, Information and Background from the Original Question to the Unclear Question.**

Figure 4: The Prompt to Construct Missing Premises Problems and Reason of Incompleteness

# Information

{information}

# Background

{background}

# Instruction

Assemble the Information and Background into a coherent text.

Generate your output STRICTLY in the following format.
Assembled Text:

# Requirements

1. **The Assembled Text SHOULD NOT CONTAIN the title of Information and Background.**
1. **The Assembled Text SHOULD CONTAIN ALL contents in the Information and Background (if the Background is not None).**
2. **DO NOT ADD ANY other content that is not included in the Information and Background to the Assembled Text.**

Figure 5: The Prompt to Construct Missing Goal Problems and Reason of Incompleteness

# Task

{query}

# Reason of Unclearness

{reason_of_unclearness}

# Instruction

According to the reason of unclearness, the task is classified to be unclear.
Please act as an impartial judge to evaluate whether the task is truly unclear and whether the reason of unclearness is correct.
If you agree that task is truly unclear, provide at least two different understandings of the unclear task corresponding to the unclearness as the verification of your judgement.

Output your final verdict by strictly following this format:
Analysis: [Your analysis about the task and the reason of unclearness]
Judge: "[[Yes]]" if you agree with the reason and that the task is truly unclear, "[[No]]" if you do not agree with the reason and that the task is clear.
Verification: [At least two different understandings of the unclear task corresponding to the unclearness]

Figure 6: The Prompt to verify the incompleteness

## B.2 HUMAN EVALUATION & QUALITY CONTROL

To verify the incomplete problems, we recruit three PhD students as human evaluators, two of them major in computer science and the other majors in mathematics. Since the incomplete problems have already been verified by Deepseek R1, we sample 10% of the problems and ask human evaluators to provide confirmation. Specifically, we ask human evaluators to provide their decision that whether the problem is incomplete given the incomplete problem, the counterpart well-defined problem, the reason of incompleteness, and R1's verification. We type the above information into Microsoft Word and present to human evaluators for them to verifiy. Totally, 128 incomplete problems are evaluated. All evaluators are paid at least the minimum wage in their country.

The evaluation results show that human agree with the R1's verification in 95% cases (i.e., human disagrees in only 5 cases). We present two cases that we provide human evaluators to make decisions. We highlight the missing elements in red.

**Potential risks of Human Evaluation**: Our human evaluation involves verification specific characteristic (i.e. the incompleteness) of mathematical problems. After a thorough examination, we do not notice any potential risk of human subjects. We have resorted to the Institutional Review Board and got the approval regarding our human evaluation. If we notice or receive any risk regarding the human evaluation, we will update this section and provide comprehensive discussion.

---

**Incomplete Problem (Missing Premises)**

---

A robot moving forward at a constant speed takes 2.5 hours to travel 1 kilometer. How many meters long is the hallway?

---

**Well-defined Problem**

---

a robot moving forward at a constant speed takes 2.5 hours to travel 1 kilometer. moving forward at this same constant speed, it takes the robot 90 seconds to travel the length of a particular hallway. how many meters long is the hallway?

---

**Reason of Incompleteness**

---

The question does not provide the time it takes for the robot to travel the hallway. While the robot's speed can be calculated from the given information (2.5 hours for 1 kilometer), the hallway's length cannot be determined without knowing how long the robot took to traverse the hallway. The missing information (e.g., 90 seconds in the Original Question) is critical for calculating the distance.

---

**R1 Verification**

---

Analysis: The task states the robot's speed (1 km in 2.5 hours) but does not specify the time it took to traverse the hallway. To calculate the hallway's length, both speed and time are required. The absence of the traversal time makes the problem unsolvable as posed. The reason for unclearness correctly identifies the missing critical information (time to traverse the hallway), rendering the task ambiguous.

Judge: [[Yes]]

Verification:
1. **Misinterpretation 1**: Assume the 2.5 hours mentioned is the time taken to traverse the hallway. This would incorrectly equate the hallway's length to 1 km (1000 meters), conflating the speed-calibration distance with the hallway's actual length.
2. **Misinterpretation 2**: Assume the problem implicitly asks for the robot's speed in meters per second (e.g., converting 1 km/2.5 hours to $\approx 0.111$ m/s) and answers with the speed instead of the hallway length, misaligning with the question's intent due to confusion over missing variables.

---

**Incomplete Problem (Missing Goals)**

In a school of 360 students, 15 are enrolled in calculus, physics, and chemistry simultaneously, while 15 students do not take any of these three subjects. A total of 180 students take calculus. The number of students taking chemistry is twice the number taking physics. There are 75 students enrolled in both calculus and chemistry, and another 75 in both physics and chemistry. However, only 30 students take both physics and calculus.

**Well-defined Problem**

there are 360 people in my school. 15 take calculus, physics, and chemistry, and 15 don't take any of them. 180 take calculus. twice as many students take chemistry as take physics. 75 take both calculus and chemistry, and 75 take both physics and chemistry. only 30 take both physics and calculus. how many students take physics?

**Reason of Incompleteness**

The problem lacks a specified goal to answer.

**R1 Verification**

Analysis: The task provides data about student enrollments in three subjects but does not specify a question to answer. The reason for unclearness—lack of a goal—is valid. Without a defined objective (e.g., finding the number of students in a specific subject or verifying consistency), the problem is ambiguous.

Judge: [[Yes]]

Verification:
1. **Understanding 1**: The task aims to find the total number of students enrolled in physics. Using the given overlaps and relationships (chemistry = 2 × physics), one might solve for physics using set theory formulas.
2. **Understanding 2**: The task requires checking if the provided numbers are mathematically consistent (e.g., ensuring overlaps do not exceed subject totals). For instance, verifying if 75 students in both physics and chemistry (with 15 in all three) do not exceed the total physics or chemistry enrollments.

## C   IMPLEMENTATION OF LLM-AS-A-JUDGE FOR EVALUATION

In this section, we detail the prompts we used to conduct LLM-as-a-Judge in Evaluation.

---

**Prompts to Evaluate that Whether a response Raises Clarification Questions when using "Implicit Prompt"**

---

# Question

{problem}

# Response

{response}

# Instruction

Please act as an impartial judge to evaluate whether the response has raised questions to ask additional information for answering the Question.

Output your final verdict by strictly following this format:
Analysis: [Your analysis about the response]
Judge: "[[Yes]]" if the response has raised questions to ask additional information, and "[[No]]" if the response directly provide an answer without raising clarification.

---

**Prompts to Evaluate CNR, i.e. whether the thoughts have noticed the need to raise clarification questions.**

---

# Incomplete Question

{incomplete_task}

# Reason of Incompleteness

{reason_of_incompleteness}

# Output

{output}

# Instruction

Please act as an impartial judge to evaluate whether the output has noticed the need to ask clarification questions to address the incompleteness.
Your assessment should be grounded on the Incomplete Question and the Reason of Incompleteness.

Output your final verdict by strictly following this format:
Analysis: [Your analysis about the Output]
Judge: "[[Yes]]" if the output has noticed the need to ask clarification questions, otherwise "[[No]]"

---

**Prompts to Evaluate ROR on missing premises problems, i.e. whether the reflection thoughts have focused on addressing the incompleteness.**

# Incomplete Question

{incomplete_task}

# Reason of Incompleteness

{reason_of_incompleteness}

# Output

{output}

# Instruction

Please act as an impartial judge to evaluate whether the output has tried to address the incompleteness.
For example, the output tried different possibilities about the incompleteness or guessing how the incompleteness arises. Your assessment should be grounded on the Incomplete Question and the Reason of Incompleteness.

Output your final verdict by strictly following this format:
Analysis: [Your analysis about the Output]
Judge: "[[Yes]]" if the output has tried to address the incompleteness, otherwise "[[No]]"

---

**Prompts to Evaluate ROR on missing goal problems, i.e. whether the thoughts imagine a goal by themselves.**

# Incomplete Question

{unclear_task}

# Reason of Incompleteness

{reason_of_unclearness}

# Output

{output}

# Instruction

According to the reason of incompleteness, the question lacks a specific goal.
Please act as an impartial judge to evaluate whether the output has tried to address the incompleteness by imagining a goal itselves.

Output your final verdict by strictly following this format: Analysis: [Your analysis about the Output]
Judge: "[[Yes]]" if the output has tried to address the incompleteness, otherwise "[[No]]"

# D  ADDITIONAL EXPERIMENTAL ANALYSIS

## D.1  ADDITIONAL ANALYSIS FOR RQ1

### D.1.1  ADDITIONAL RESULTS ON DATASET SYNTHESIZED BY GEMINI-2.5 PRO

Table 7:  Coarse-grained evaluation of LRMs on datasets synthesized by Gemini-2.5 Pro.

| Models | Missing Premises | | | Missing Goal | | |
|---|---|---|---|---|---|---|
| | CR | TLC | TLNC | CR | TLC | TLNC |
| CRITIC-math (from Math 500 Data Source) | | | | | | |
| Deepseek R1 | 57.00% | 0.61 | 4.03 | 52.00% | 0.88 | 2.56 |
| Qwen3 Plus | 53.00% | 0.33 | 2.38 | **55.98%** | 0.27 | 0.85 |
| o3 mini | 55.00% | 1.50 | 2.12 | 34.00% | 1.42 | 1.18 |
| Grok 3 mini | **58.00%** | 0.50 | 1.43 | 47.00% | 0.47 | 1.16 |
| Claude 3.7 | 48.00% | 0.76 | 2.28 | 40.00% | 0.66 | 2.06 |
| CRITIC-math (from Omni-Math Data Source) | | | | | | |
| Deepseek R1 | 56.00% | 0.34 | 2.38 | 55.00% | 0.40 | 1.34 |
| Qwen3 Plus | 45.00% | 0.63 | 1.54 | **57.09%** | 0.41 | 1.12 |
| o3 mini | 47.00% | 0.52 | 1.19 | 45.00% | 0.45 | 0.88 |
| Grok 3 mini | **52.00%** | 0.33 | 1.00 | 52.00% | 0.37 | 0.96 |
| Claude 3.7 | 44.00% | 0.92 | 2.14 | 35.00% | 0.33 | 1.73 |

To avoid model bias in data synthesis, we additionally use Google's Gemini 2.5 Pro to generate supplementary data and perform corresponding evaluation. As detailed in the table, our core findings consistently hold across data synthesized by different models. These include:

• LRMs perform poor to ask for information on incomplete problems (low CR-Clarification Ratio).

• LRMs tend to overthink on incomplete problems (large TLNC-Thinking Length Not Clarifying).

• Raising clarification questions on incomplete problems allow LRMs to reduce thinking effort (low TLC, Thinking Length when Clarifying).

### D.1.2  ADDITIONAL RESULTS USING GPT-4.1 AS THE JUDGE LLM

Table 8:  Fine-grained analysis of LRMs with GPT-4.1 as the Judge LLM.

| Model | Missing Premises | | Missing Goal | |
|---|---|---|---|---|
| | ROR | CNR | ROR | CNR |
| Deepseek R1 | 75.62% | 40.63% | 84.45% | 16.39% |
| Qwen3 Plus | 77.38% | 45.71% | 80.19% | 16.51% |

Table 9:  Fine-grained analysis of LRMs with Claude-3.7 Thinking as the Judge LLM.

| Model | Missing Premises | | Missing Goal | |
|---|---|---|---|---|
| | ROR | CNR | ROR | CNR |
| Deepseek R1 | 79.91% | 58.69% | 98.74% | 35.29% |
| Qwen3 Plus | 79.29% | 61.90% | 97.64% | 4.81% |

We provide additional results where we use other two LLMs to judge the thoughts of Deepseek R1 and Qwen3 Plus. As detailed in the table, our main findings remain consistent with those in our main page, even when evaluated by different LLMs. Specifically, the high ratio of overthinking and high CNR (Clarification Noticing Ratio) in scenarios with missing information are consistently observed across these different evaluators, although the specific evaluation scores may vary between models.

### D.1.3 THE RELATIONSHIP BETWEEN CRs AND PROBLEM DIFFICULTIES

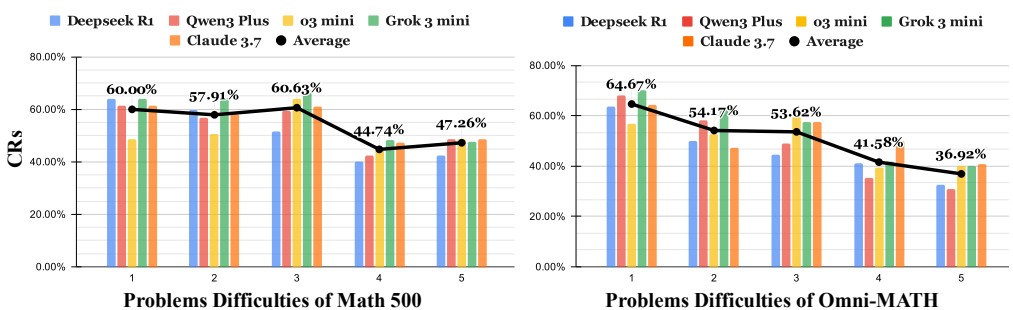

Figure 7: CRs of different LRMs on missing premises problems with different difficulties.

Figure 7 show the clarification ratios (CRs) of different LRMs on missing premises problems. We can observe a negetive correlation between the problem difficulty and the CRs, which indicate that the ability to ask for information is unrobust to the prolem difficulty.

### D.1.4 QUESTIONS ACCURACIES IN ADDRESSING THE INCOMPLETENESS

In this section, we provide the evaluation results regarding the accuracy of asking for information, which is measured by the ratio of questions asked by LRMs that can appropriately address the incompleteness. We use LLM-as-a-Judge (Deepseek R1) as the evaluation, where the prompt are detailed in Figure 8. From the results, we can observe that all LRMs, when they successfully raise questions, can achieve rather high accuracy in asking for information.

Table 10: The accuracy of asking for information, which is measured by the ratio of questions asked by LRMs that can appropriately address the incompleteness.

| Model | Missing Premises | Missing Goal |
|---|---|---|
| CRITIC-math (from Math 500 Data Source) | | |
| Deepseek R1 | 91.25% | 98.25% |
| Qwen3 Plus | 95.41% | 99.26% |
| o3 mini | 89.16% | 90.48% |
| Grok 3 mini | 92.23% | 100.00% |
| Claude 3.7 | 87.26% | 99.12% |
| CRITIC-math (from Omni-MATH Data Source) | | |
| Deepseek R1 | 87.18% | 95.59% |
| Qwen3 Plus | 89.86% | 95.92% |
| o3 mini | 87.00% | 91.53% |
| Grok 3 mini | 85.51% | 98.04% |
| Claude 3.7 | 85.53% | 95.71% |

### D.1.5 SAMPLING MULTIPLE RESPONSES

. In the evaluation in Section 3, we generate one response with the temperature set to 0 from close-source LRMs. To analyze that how the temperature affects the results and will the results significantly vary in different samples on the same problem, we conduct additional evaluation. Considering our budget, we conduct this evaluation only using the "explicit prompt" on randomly sampled 50 problems from each category of missing premises, missing goal, and well-defined in each data source.

```
# Incomplete Task

{incomplete_task}

# Reason of Incompleteness

{reason_of_incompleteness}

# Clarification Question

{clarification_question}

# Instruction

Please act as an impartial judge to evaluate whether the clarification question can
appropriately address the incompleteness of the incomplete question, given the content of
the incomplete question and the reason of incompleteness.

Output your final verdict by strictly following this format:
Analysis: [Your analysis about the clarification question]
Judge: "[[Yes]]" if the task clarification question is appropriate and sufficient, otherwise
"[[No]]"
```

Figure 8: Prompts to evaluate the accuracy of asking for information.

For each problem, we generate one response with temperature set to 0 (denotes as S) and 5 responses
with temperature set to 0.6 (denoted as M).

Table 11 display the results. From the results, we can observe that sampling multiple responses will
slightly influence the CRs, TLCs, and TLNCs. However, the influence varies in different LRMs
and different types of incompleteness. Generally speaking, the results of S mostly fall into the 95%
confidential interval of M. Also, the main conclusion remains the same, where the CRs are around
50%, TLNCs are greater than 1 on missing premises problems, barely changed TLNCs on missing
goal problems, and lower TLCs on both cases. These additional results demonstrate that our analysis
and results in the main text are robust to the choice of temperature and different sampled responses.

**TLNCs on Problems of Different Difficulties**.

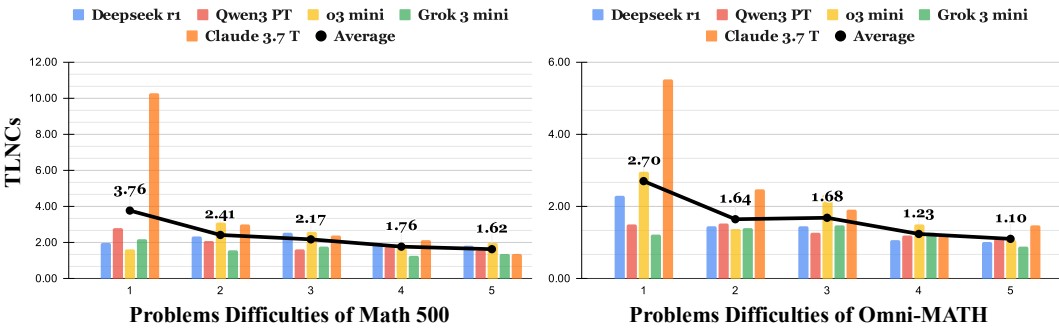

Figure 9: TLNCs of different LRMs on incomplete problems with different difficulties.

The results are plotted in Figure 9. From the results, we can observe a negative correlation between the
problem difficulty and the TLNCs, where the overthink are more significant on easier problems. For
example, on the difficulty level of 1 in Math 500, the thoughts lengths are in average 3.76 times longer
thoughts compared with the lengths on well-defined problems. In addition, the thoughts lengths are

Table 11: The results of sampling multiple responses. S denotes generating one response with temperature set to 0, and M denotes sampling 5 responses with temperature set to 0.6. For M, 95% confidential intervals are reported. We omit the results of well-defined problems for better readability.

| Model | Type | Missing Premises | | | Missing Goal | | |
|---|---|---|---|---|---|---|---|
| | | CR | TLC | TLNC | CR | TLC | TLNC |
| CRITIC-math (Subset from Math 500 Data Source) | | | | | | | |
| Deepseek R1 | S | 55.56% | 0.53 | 2.25 | 46.00% | 0.41 | 1.03 |
| | M | 37.60%±3.24% | 0.62±0.32 | 2.38±0.22 | 48.80%±4.84% | 0.49±0.13 | 1.29±0.22 |
| Qwen3 PT | S | 44.00% | 0.50 | 2.45 | 64.00% | 0.73 | 1.23 |
| | M | 44.00%±3.51% | 0.55±0.01 | 2.60±0.13 | 60.00%±2.51% | 0.47±0.01 | 1.31±0.26 |
| o3 mini | S | 46.00% | 1.62 | 3.20 | 36.00% | 1.54 | 1.33 |
| | M | 43.20%±1.36% | 1.78±0.37 | 2.98±0.65 | 38.80%±0.02 | 1.88±0.33 | 1.32±0.58 |
| Grok 3 mini | S | 46.00% | 0.57 | 1.61 | 54.00% | 0.64 | 0.86 |
| | M | 48.00%±3.04% | 0.70±0.19 | 1.72±0.20 | 49.20%±2.83% | 0.47±0.04 | 1.15±0.09 |
| Claude 3.7 | S | 28.00% | 0.30 | 2.56 | 46.00% | 0.16 | 1.59 |
| | M | 38.40%±17.86% | 0.29±0.24 | 1.76±1.12 | 48.00%±8.42% | 0.28±0.27 | 1.51±0.74 |
| CRITIC-math (Subset from Omni-Math Data Source) | | | | | | | |
| Deepseek R1 | S | 46% | 0.28 | 1.25 | 59.00% | 0.24 | 1.16 |
| | M | 50.00%±3.51% | 0.39±0.03 | 1.28±0.08 | 50.80%±0.42 | 0.26±0.12 | 1.33±0.09 |
| Qwen3 PT | S | 48.00% | 0.23 | 1.24 | 58.00% | 0.27 | 1.26 |
| | M | 51.20%±2.22% | 0.33±0.03 | 1.21±0.08 | 58.00%±4.65% | 0.33±0.07 | 1.21±0.14 |
| o3 mini | S | 56.00% | 0.42 | 1.71 | 40.00% | 0.72 | 1.08 |
| | M | 55.60%±6.43% | 0.57±0.39 | 1.48±0.14 | 39.20%±0.03 | 0.66±0.13 | 0.87±0.13 |
| Grok 3 mini | S | 56.00% | 0.44 | 1.07 | 52.00% | 0.45 | 1.10 |
| | M | 61.60%±5.38% | 0.40±0.05 | 1.06±0.09 | 53.60%±3.24% | 0.41±0.04 | 1.07±0.10 |
| Claude 3.7 | S | 42.00% | 0.45 | 1.38 | 38.00% | 0.36 | 1.16 |
| | M | 42.00%±1.76% | 0.45±0.14 | 1.45±0.10 | 36.40%±4.44% | 0.21±0.11 | 1.27±0.14 |

increased more significant on Math 500 compared with Omni-MATH, given that Omni-MATH is a dataset containing more challenging problems.

### D.1.6 CASES STUDIES ABOUT OVERTHINKING WHEN MISSING PREMISES

We provide 3 case studies of Deepseek R1 with "explicit prompt" related to the overthinking when missing premises. The case studies are provided in Figure 10, 11, and 12. From the cases, we can observe that LRMs try to address the incompleteness by themselves (marked as **red**) rather than asking for information. Such self-reflections on incompleteness suggests that the current mode of deep-thinking may overemphasize the internal thinking while ignoring the external interactions. These phenomena also support our analysis in Section D.2 about the dilemma between learning the current mode of deep-thinking and asking for information.

### D.1.7 CASES STUDIES ABOUT HALLUCINATIONS WHEN MISSING GOALS

We provide 6 case studies of Deepseek R1 (3 with "implicit prompt" and 3 with "explicit prompt") about the hallucinations when missing goals. The results are shown in Figure 13, 14, 15, 16, 17, and 18. We highlight the goal that not specified in the input but imagined by Deepseek R1 in **red**. In those cases, we can observe that Deepseek R1 immediately generates a goal in their first paragraph of thinking, and dedicates to solve the imagined goal and finally provide a corresponding answer.

# Problem to Solve

The greek army contained two types of soldiers: the upper class and the lower class soldiers. If there were a total of 5 upper class soldiers and 10 lower class soldiers in a certain part of athens, and the battle of thermopylae demands a force of 4 upper class soldiers, how many different battalions can be sent?

# Instruction

If you need ask information, please raise clarification question and start your response STRICTLY with: "Clarification Question" followed by your questions.
Otherwise, please reason step by step, and put your final answer within \{}.

...

**But wait, maybe there's more to it. The problem mentions "battalions," which usually consist of multiple soldiers.** However, since the requirement is specifically for 4 upper class soldiers, perhaps lower class soldiers are not part of this battalion. Alternatively, maybe the problem expects that the battalion includes both upper and lower class soldiers, but with exactly 4 upper class ones. But the problem doesn't specify a total number of soldiers needed for the battalion. So that's a bit confusing.

...

**Alternatively, maybe the problem is just asking for the number of ways to choose the 4 upper class soldiers from the available 5**, regardless of the lower class soldiers. Because if the battalion is required to have 4 upper class soldiers, and the number of lower class soldiers isn't specified,

...

**But I need to make sure. Let me think again. The problem states "a force of 4 upper class soldiers."** If the battalion is just those 4, then yes, 5 choose 4 is 5.

...

Figure 10: Case Studies of Overthinking when Missing Premises.

# Problem to Solve

find the remainder when $(n + 2)(n + 4)(n + 6)$ is divided by 7.

# Instruction

If you need ask information, please raise clarification question and start your response STRICTLY with: "Clarification Question" followed by your questions.
Otherwise, please reason step by step, and put your final answer within \{}.

---

...
**Case 1: n = 0 mod 7**
Then:
n + 2 = 2 mod 7
n + 4 = 4 mod 7
n + 6 = 6 mod 7
Product: 2*4*6 = 48. 48 divided by 7 is 6*7=42, remainder 6. So 48 mod7=6.

...
**Case 2: n =1 mod7**
Then:
n+2 =3 mod7
n+4=5 mod7
n+6=7=0 mod7
Product: 3*5*0=0. So remainder 0.

...
But wait, the problem is asking for "the remainder" in general. But since the remainder depends on n, unless there's more information about n, we can't give a unique answer. **Wait, but maybe the problem is expecting an answer that works for any n, or perhaps there's a pattern that can be generalized**.
...

Figure 11: Case Studies of Overthinking when Missing Premises.

# Problem to Solve

a pentagon is rotated counterclockwise. what is the number of degrees it must be rotated until it coincides with its original position?

# Instruction

If you need ask information, please raise clarification question and start your response STRICTLY with: "Clarification Question" followed by your questions.
Otherwise, please reason step by step, and put your final answer within \{}.

---

...
The exact wording is: "what is the number of degrees it must be rotated until it coincides with its original position?" So, does that mean the total rotation needed to get back to the original position? For example, if you rotate 72 degrees each time, after five such rotations, you would have 5*72 = 360 degrees, which brings it back. **But is the question asking for the smallest such angle?** Because 360 degrees is a full rotation, which would obviously bring it back. But maybe the question is trickier than that
But I need to make sure. Let me think. For example, a square has rotational symmetry every 90 degrees. So, rotating it 90 degrees four times brings it back to the original position. So, each 90 degrees is a step. So, the minimal angle is 90 degrees. But the total rotation required to get back to original is 360 degrees, but the question is probably asking for the minimal angle that when you rotate by that angle, the figure coincides with itself.
...

Figure 12: Case Studies of Overthinking when Missing Premises.

---

A proper divisor of an integer n is a positive divisor of n that is less than n. For example, the proper divisors of 12 are 1, 2, 3, 4, and 6.

---

Okay, **so I need to figure out the proper divisors of an integer**. Let's start by recalling what a proper divisor is. The problem statement says a proper divisor of an integer n is a positive divisor that is less than n. For example, for 12, the proper divisors are 1, 2, 3, 4, and 6. That makes sense because those numbers divide 12 without leaving a remainder and are all less than 12.
...
**The proper divisors of 12 are** $\{1, 2, 3, 4, 6\}$ .

Figure 13: Case Studies of Hallucinations when Missing Goals.

---

$\overline{bc}$ is parallel to the segment through $a$, and $ab = bc$. The diagram shows two horizontal lines connected by segments from $a$ to $b$ and $a$ to $c$. The angle at $a$ is labeled $124°$, and an angle labeled $x°$ is marked on the upper horizontal line near $(4.5, 3)$.

---

Okay, let's see. **I need to figure out the value of angle x in this diagram**. Hmm. The problem says that segment bc is parallel to the segment through a, and that ab equals bc. There are two horizontal lines connected by segments from a to b and a to c. The angle at a is 124 degrees, and angle x is on the upper horizontal line near the point (4.5, 3).
...
**Final Answer** 28

Figure 14: Case Studies of Hallucinations when Missing Goals.

The problem must be solved without the use of a calculator.

Okay, so I need to solve this problem without using a calculator. **Let me see... The problem is about a triangle with vertices at points A, B, and C**. The coordinates given are A(1, 2), B(4, 6), and C(6, 3). The question is asking for the area of this triangle. Hmm, right. I remember there are a few ways to calculate the area of a triangle when you have the coordinates of the vertices. One method is using the shoelace formula, another might be using vectors or determinants. Let me recall the shoelace formula because that seems straightforward for coordinates.

...

**The area of the triangle is** $\boxed{\dfrac{17}{2}}$.

Figure 15: Case Studies of Hallucinations when Missing Goals.

# Problem to Solve

A regular hexagon can be divided into six equilateral triangles. The perimeter of one of the triangles is 21 inches.

# Instruction

If you need ask information, please raise clarification question and start your response STRICTLY with: "Clarification Question" followed by your questions.
Otherwise, please reason step by step, and put your final answer within \boxed{}.

Okay, let's see. The problem says a regular hexagon can be divided into six equilateral triangles. The perimeter of one of those triangles is 21 inches. I need to find something, but wait, the original problem statement isn't here. Wait, looking back. **Oh, the user said "If you need to ask information, please raise clarification question..." So maybe the problem is to find the perimeter of the hexagon? Because that would make sense.**

...

So yes, the side of each triangle is equal to the side of the hexagon. **Therefore, perimeter of hexagon is 6 * 7 = 42. So I think that's correct**

Figure 16: Case Studies of Hallucinations when Missing Goals.

# Problem to Solve

Let $z = 2 + \sqrt{2} - (3 + 3\sqrt{2})i$ and $c = 2 - 3i$. The point $w$ is obtained by rotating $z$ around $c$ by $\frac{\pi}{4}$ radians counter-clockwise. An Asymptote diagram illustrates this configuration: $z$ is plotted at the coordinate $(2 + \sqrt{2}, -3 - 3\sqrt{2})$, $c$ at $(2, -3)$, and $w$ is the result of rotating $z$ by $45°$ around $c$. The figure includes lines connecting $z$, $c$, and $w$, with labeled points and an angle label $\frac{\pi}{4}$ near $c$.

# Instruction

If you need ask information, please raise clarification question and start your response STRICTLY with: "Clarification Question" followed by your questions.
Otherwise, please reason step by step, and put your final answer within \boxed{}.

---

Okay, **so I need to find the point w**, which is the result of rotating z around c by $\pi/4$ radians counter-clockwise. Let me start by recalling how rotation works in the complex plane.
...
Therefore, **the complex number w is 6 -5i**. So I think that's the correct answer. I don't see any mistakes in the steps. So the final answer is $\boxed{6 - 5i}$.

Figure 17: Case Studies of Hallucinations when Missing Goals.

# Problem to Solve

Denali and Nate work for a dog walking business and are paid for each dog they walk. Denali is responsible for 16 dogs, and Nate is responsible for 12 dogs. Under the company's new policy, they will be assigned or unassigned new dogs in groups of $x$ dogs. The ratio of Denali's pay to Nate's pay would be the same if Denali started walking $4x$ more dogs and Nate stayed at 12 dogs or if $x$ of Nate's dogs were reassigned to Denali.

# Instruction

If you need ask information, please raise clarification question and start your response STRICTLY with: "Clarification Question" followed by your questions.
Otherwise, please reason step by step, and put your final answer within \boxed{}.

---

Okay, let's see. So the problem is about Denali and Nate who work for a dog walking business. They get paid per dog they walk. Currently, Denali has 16 dogs and Nate has 12. The company is changing their policy so that they'll be assigned or unassigned dogs in groups of x dogs. The key point is that the ratio of their pay remains the same under two different scenarios: either Denali gets 4x more dogs and Nate stays at 12, or x of Nate's dogs are reassigned to Denali. **I need to find x**. ... Yes, both equal 3. So the answer is 5. So x=5.
**Therefore, boxed answer is** $\boxed{5}$

Figure 18: Case Studies of Hallucinations when Missing Goals.

## D.2 Additional Analysis for RQ2

We conduct ablation studies to analyze the effect of different typs of problems in SFT models. The results are provided in Table 12, where the subscript W indicates trained solely using well-defined problems, and the subscript I indicates trained solely using incomplete problems. From the results, we have the following observations:

**Learning to ask for information can benefit the ability of solving problems**. Comparing CRITIC-Qwen$_W$ with CRITIC-Qwen and CRITIC-Qwen-thinking$_W$ with CRITIC-Qwen-thinking, we can observe that SRs on well-defined problems even decreased when solely learning to solve well-defined problems (W). Therefore, learning to ask for information on incomplete problems is not contradict with, even benefit, learning the ability to solve well-defined problems

**The Dilemma between Deep Thinking and Asking for Information**. When trained solely on the answers of incomplete problems, CRITIC-Qwen$_I$ achieves 100% CR on both types of problems, which makes sense since the SFT model will overfit to raising clarification questions. However, when trained with the thoughts (CRITIC-Qwen-thinking$_I$), the CRs significantly decrease, and comparable SRs are achieved on well-defined problems where SFT models do not raise questions. This suggests that the current mode of thoughts overemphasize the contents regarding how to solve problems, even when thinking to raise clarification questions. Trained on such kind of deep thinking process, the model will more likely to just learn how to solve problems rather than how to interact with users and ask for information. In addition, we can observe that CRITIC-Qwen-thinking$_I$ achieve higher CRs compared with CRITIC-Qwen-thinking where their deep thinking is less strengthened by learning to solve well-defined problems, therefore being more willing to ask for information.

Table 12: Ablation studies of SFT models trained on different types of problems.

| Model | Missing Premises | Missing Goal | Well-defined | |
|---|---|---|---|---|
| | CR | CR | CR | SR |
| CRITIC-math (from Math 500 Data Source) | | | | |
| CRITIC-Qwen | 78.42% | 94.87% | 4.12% | 73.39% |
| CRITIC-Qwen$_W$ | 0.53% | 0.00% | 0.00% | 73.25% |
| CRITIC-Qwen$_I$ | 100.00% | 100.00% | 100.00% | 0.00% |
| CRITIC-Qwen-thinking | 57.37% | 62.82% | 1.23% | 80.83% |
| CRITIC-Qwen-thinking$_W$ | 34.21% | 28.63% | 0.00% | 80.25% |
| CRITIC-Qwen-thinking$_I$ | 63.42% | 73.93% | 7.00% | 79.20% |
| CRITIC-math (from Omni-MATH Data Source) | | | | |
| CRITIC-Qwen | 77.88% | 97.64% | 13.16% | 41.13% |
| CRITIC-Qwen$_W$ | 0.23% | 0.00% | 0.00% | 39.85% |
| CRITIC-Qwen$_I$ | 100.00% | 100.00% | 100.00% | 0.00% |
| CRITIC-Qwen-thinking | 56.66% | 69.29% | 7.52% | 54.88% |
| CRITIC-Qwen-thinking$_W$ | 24.60% | 37.40% | 3.38% | 51.75% |
| CRITIC-Qwen-thinking$_I$ | 57.34% | 69.53% | 12.41% | 52.79% |

In Figure 19 we provide a case study about the training samples. We can observe that, even the final answer raises clarification questions and LRMs quickly identify the missing information (marked in **blue**), the thinking process still tries hard to address the incompleteness (marked in **red**). Such thinking process strengthen the ability of solving problems (i.e., trying different ways to solve a problem) while weaken the ability of asking for information.

### D.2.1 Additional Results trained using Llama-3.1-8B-Instruct

We have further fine-tune Llama 3.1 8B Instruct to analyze whether "deep-thinking dillemma" exhibits across different models. The results are shown in the above table. As detailed in the table, our core findings regarding "deep-thinking dillemma" consistently hold when fine-tuning Llama 3.1 8B Instruct, an instruction-tuned model as you requested. These include:

> # Problem to Solve
>
> a group of boys and girls have eaten at a restaurant where only pizzas cut into 12 slices are served. each boy ate 6 or 7 slices and each girl 2 or 3 slices. calculate the number of boys and girls in the group.
>
> # Instruction
>
> If you need ask information, please raise clarification question and start your response STRICTLY with: "Clarification Question" followed by your questions.
> Otherwise, please reason step by step, and put your final answer within \boxed{}.
>
> ---
>
> <think>
> Okay, let's try to figure out this problem. So, there are some boys and girls who ate pizza at a restaurant. The pizzas are each cut into 12 slices. Each boy ate either 6 or 7 slices, and each girl ate 2 or 3 slices. We need to find out how many boys and girls are in the group.
>
> Hmm, the problem says "calculate the number of boys and girls," but I don't see any specific numbers given here. **Maybe there's some missing information? Wait, maybe the total number of slices or the number of pizzas ordered is implied? Let me check the problem again.**
>
> Wait, the problem states that "a group of boys and girls have eaten at a restaurant where only pizzas cut into 12 slices are served." So the total number of slices must be a multiple of 12 because they can only order whole pizzas. But the problem doesn't mention how many pizzas they ordered. That's probably the key here. So the total slices consumed by the group must be a multiple of 12.
>
> **Let me restate the problem.** Let's denote the number of boys as B and girls as G. Each boy ate 6 or 7 slices, so total slices eaten by boys would be 6B or 7B, but actually, each individual boy can eat either 6 or 7. So the total for boys is 6B + some number depending on how many ate 7 instead of 6. Similarly, each girl ate 2 or 3, so total for girls is 2G + some number depending on how many ate 3. But this seems complicated because there are multiple variables and possible combinations.
>
> **Wait, maybe there's more constraints.** The problem might be expecting a unique answer, which would mean that the total number of slices is a multiple of 12, and with the number of boys and girls such that the possible combinations of 6/7 and 2/3 sum up to that multiple. But without knowing the total slices or the number of pizzas, how can we find a unique answer?
> ...
> So I need to ask for clarification. The problem must have had more data, but maybe it's missing in translation. Therefore, the correct approach here is to ask for the total number of pizzas or total slices consumed.

Figure 19: Case Studies of One Training Sample.

Table 13: Coarse level evaluation of SFT models. *CRITIC-Qwen T* denotes CRITIC-Qwen-thinking. *Qwen3-8B T* denotes the official LRM (also fine-tuned from Qwen3-8B-Base) with thinking enabled.

| Model | Missing Premises | | | Missing Goal | | | Well-Defined | | | | ACC |
|---|---|---|---|---|---|---|---|---|---|---|---|
| | CR | TLC | TLNC | CR | TLC | TLNC | CR | TLC | TLNC | SR | |
| CRITIC-Llama | **88.42%** | / | / | **90.15%** | / | / | 27.12% | / | / | 32.10% | **88.81%** |
| CRITIC-Llama T | 66.32% | 0.11 | 1.50 | 73.93% | 0.06 | 1.02 | 7.82% | 0.34 | 1 (4554) | 43.62% | 77.48% |
| Llama-3.1-8B-Instruct | 51.30% | / | / | 49.57% | / | / | 41.98% | / | / | **6.17%** | 64.47% |

- Fine-tuning on training data of CRITIC helps to improve the ability of asking for information of Llama, suggested by the improved clarification accuracy (ACC: the ratio of clarifying incomplete questions but not clarifying well-defined questions) of CRITIC-Llama compared with Llama 3.1 8B Instruct.

- Incorporating deep-thinking, the problem-solving ability of fine-tuned model is improved, suggested by the higher SR (solving ratio of well-defined problems) of CRITIC-Llama T(hinking) compared with CRITIC-Llama.

- Incorporating deep-thinking, the asking for information ability of fine-tuned model is reduced, suggested by the lower CR (clarification ratio) and ACC (clarification accuracy) of CRITIC-Llama T compared with CRITIC-Llama.

The above results exactly demonstrate the deep-thinking dilemma, where incorporating current style of deep-thinking improves the problem-solving ability while reduces the ability to ask for information. Such results alert the community that the current type of thinking may overlook another ability of genuine intelligence.

# E   LLMs Usage Statement

LLMs/LRMs become powerful tools for efficiently processing data and human-like evaluation in many recent works. Following these common practices, in this paper, we use LLMs to synthesize large-scale incomplete problems and conduct LLM-as-a-Judge evaluation. However, all conceptual work and method design in this work are conducted independently by the authors. LLMs are used to polish the writing, help to synthesize data, and evaluate results, which do not contribute to the dataset design or the shapre of other core ideas of this paper. Any part steps that involve LLMs to synthesize data and evaluate results are guided by carefully designed frameworks proposed by the authors.

