# OpenReview forum: "Beyond Solving Math Quiz: Evaluating the Ability of Large Reasoning Models to Ask for Information"
_ICLR.cc/2026/Conference — ICLR 2026 Conference Withdrawn Submission_

### Official Review · Reviewer_JV3H · 2025-10-28

**Soundness:** 2
**Presentation:** 3
**Contribution:** 2
**Rating:** 4
**Confidence:** 4

**Summary:**

Against the limitation that existing Large Reasoning Models (LRMs) are only evaluated on solving well-defined mathematical problems, this paper proposes the CRITIC-math benchmark dataset covering two types of incomplete mathematical problems, namely "Missing Goal" and "Missing Premises", and points out that current LRMs are insufficient in proactively asking for information.

**Strengths:**

1. The paper is clearly expressed, with well-defined motivations and solutions, making it easy to read.
2. The paper proposes a new benchmark, which may be of certain help to subsequent work in this field.

**Weaknesses:**

1. Current large language models are not solely evaluated on solving well-defined mathematical problems. Currently, there are numerous benchmarks and discussions regarding ill-defined, pathological, and unsolvable mathematical problems, and the current difficulty lies in solving such problems.
2. The conclusion of the paper is not novel. The inadequacy of large language models in proactively asking for information has already been widely discussed in the field. So this is hardly a new finding.
3. The paper describes a phenomenon but provides little insight into how to solve the problem, resulting in limited contributions.
4. Doubts about the Rationality of Evaluation Metrics: The paper adopts metrics such as "Clarification Ratio (CR)" and "Clarification Accuracy (ACC)", but fails to clearly define the criteria for "effective clarification questions". For instance, the document mentions that "the accuracy of LRMs when asking questions is relatively high (approximately 85%-100%)", yet it does not specify whether the determination of "accuracy" takes into account the "relevance between the questions asked and the missing information".

**Questions:**

None.

---

### Official Review · Reviewer_mfWt · 2025-10-29

**Soundness:** 2
**Presentation:** 2
**Contribution:** 2
**Rating:** 4
**Confidence:** 4

**Summary:**

The paper proposes CRITIC-math, a benchmark for testing whether LRMs proactively ask for missing information instead of “solving” underspecified math problems. The dataset is built by taking well-defined problems from MATH-500 and Omni-MATH and programmatically removing either the goal or one premise, then reassembling text into free-form problems and verifying “incompleteness” with LLMs plus light human checks. The authors evaluate several closed models and show low clarification ratios unless explicitly prompted, plus analyses of “overthinking” and “hallucinating the goal.” They also fine-tune smaller open models to ask questions and report higher clarification accuracy, with a trade-off when adding long “thinking.”

**Strengths:**

- Problem framing is timely. Asking for clarification on incomplete inputs is useful, and the paper makes this failure mode concrete for math. The coarse and fine-grained metrics (CR/ACC, TLC/TLNC, RS/ROR/CNR) are clearly defined.
- Clear empirical signal. Under “implicit prompts,” clarification ratios hover around ~25–35% and only rise to ~50–60% with explicit instructions, which is a crisp takeaway.
- Readable, well-organized paper. The data construction pipeline is explained step-by-step, and the failure-mode analysis (overthinking, hallucinated goals) is easy to follow with examples.

**Weaknesses:**

- Benchmark necessity vs. saturation. We already have substantial math benchmarks (e.g., MATH (12.5k), GSM8K (8.5k), Omni-MATH (4.4k Olympiad-level)) and even multimodal math benchmarks like MathVista targeting visual reasoning. The paper’s case for “another” math dataset is not fully persuasive. Its novelty is primarily removal of goals/premises plus evaluation of question-asking. Related efforts like QuestBench study information acquisition explicitly (albeit with multiple-choice question selection), which this paper cites but doesn’t position against strongly enough. A clearer gap analysis would help.
- Data realism and construction artifacts. CRITIC-math is created by decomposing problems into “goal/premises/background,” then deleting one element and LLM-reassembling into free-form text. This risks distribution shift: the resulting “underspecified” items may reflect the style and biases of DeepSeek-R1 prompts rather than natural user underspecification. The paper uses LLMs both to construct and verify “unclearness” with limited human sampling, raising leakage and circularity concerns. Stronger human validation on a larger sample and cross-model construction would reduce this risk.
- Heavy reliance on LLM-as-a-judge. Key labels (did the model ask for clarification? did thoughts notice incompleteness?) come from LLM judges, even for fine-grained metrics. Though the authors try multiple judges, the evaluation still depends on opaque judgments. More human double-annotation, adjudication, and inter-annotator agreement would strengthen claims.
- Limited novelty over existing “clarification/ambiguity” lines. There’s a broader literature on ambiguity and clarification in NLP and dialogue, and math-specific “underspecified” settings are emerging (e.g., QuestBench includes GSM-style items with missing info). The paper would benefit from a tighter comparison of task formulations and metrics, and from showing where CRITIC-math reveals behaviors not surfaced by those prior datasets.
- External validity and scope. By design, problems are made incomplete via synthetic deletions. It’s unclear whether models that do well here will handle naturally underspecified questions from users. The paper hints at extending to programming but doesn’t show cross-domain transfer.
- Mixed story on the training takeaway. The SFT section shows a “deep-thinking vs. clarification” trade-off, which is interesting, but the setup blends multiple moving parts (teacher model, explicit prompting style, selection filters). It’s not yet a general recipe practitioners can apply, and the gains come with side effects like higher false positives on well-defined items.

**Questions:**

- Why a new math benchmark now? Can you articulate precisely what CRITIC-math evaluates that MATH, GSM8K, Omni-MATH, and MathVista cannot, and why QuestBench’s underspecification setup isn’t sufficient? A direct, table-style comparison of task definitions, label sources, and metrics would help.
- Construction bias. How do you guard against artifacts from using DeepSeek-R1 in both construction and verification? Did you repeat the entire pipeline with a different model family and re-run all results to check robustness?
- Human validation scale. What fraction of the dataset received human verification, and what was the agreement? Would you consider releasing a fully human-vetted subset for benchmark stability?
- Generalization beyond synthetic deletions. Do models trained on CRITIC-math transfer to real user incomplete math queries (e.g., forum posts) without format cues? Any small real-world test set?
- Metric sensitivity. ACC mixes “ask when incomplete” and “don’t ask when complete.” How sensitive are the conclusions to different decision thresholds or to false-positive penalties on well-defined problems?
- Practical guidance. If I’m a practitioner, what’s the recommended training recipe to encourage clarification without hurting solving ability? Can you quantify the compute/latency impact of your best-performing SFT variant?

---

### Official Review · Reviewer_py4d · 2025-11-01

**Soundness:** 2
**Presentation:** 2
**Contribution:** 2
**Rating:** 4
**Confidence:** 4

**Summary:**

This paper proposes that genuine artificial intelligence should not only solve mathematical problems but also identify when a problem lacks sufficient information and proactively ask for clarification. To study this overlooked capability, the authors introduce CRITIC-math, a new benchmark dataset of over 6,000 mathematically incomplete problems (missing goals or premises) derived from existing math datasets. Systematic evaluation of leading Large Reasoning Models (LRMs) shows that they rarely ask for missing information, instead overthinking, hallucinating goals, or attempting to “solve” ill-posed problems. Through supervised fine-tuning experiments, the authors show that models can learn to ask for information more effectively, though a trade-off (“dilemma”) emerges between deep reasoning and information-seeking behavior. The study highlights that current LRM development is overly biased toward problem-solving, calling for new approaches that foster proactive, context-aware intelligence in AI systems.

**Strengths:**

- Clear motivation. Convincingly argues that genuine mathematical intelligence requires proactively asking for missing information, not just solving well-posed problems, and anchors this in concrete real-world examples and prior perspectives on intelligence.
- The paper introduces a clear definition of incomplete mathematical problems.
- Insightful fine-grained analysis that goes beyond averages to examine thoughts, reflection steps, and noticing vs. acting on incompleteness—surfacing “thoughts-to-answer unfaithfulness” and quantifying reflection on incompleteness.

**Weaknesses:**

- The dataset is largely LLM-synthesized (DeepSeek R1 and Gemini 2.5) via templated disturbances (blanking the goal or removing a premise), potentially creating somewhat "artificial" patterns, rather than simulating how real users naturally phrase underspecified or ambiguous math questions. Human spot checks can not completely solve this.
- The key metrics depend on LLM-as-a-judge (primarily DeepSeek R1) to detect “clarification,” even while R1 is an evaluated model. This creates severe circularity and bias risks. Brief cross-checks cannot fully solve this.
- The supervised fine-tuning (SFT) process identifies positive examples using the same explicit-prompt and string-based criteria later used for evaluation. This creates a “teaching to the rubric” effect, where the model may simply learn to reproduce the evaluation trigger rather than develop genuine clarification ability. Additionally, the experiments are limited to a single 8B backbone and one-epoch training, restricting generalizability.
- The evidence for the “deep-thinking vs. asking” dilemma seems less compelling. The “CRITIC-Qwen-thinking” model’s problem-solving accuracy on MATH500 drops drastically (17%), suggesting that reduced performance might stem from training artifacts rather than a genuine cognitive trade-off.

**Questions:**

- The "Thoughts-to-Answer Unfaithfulness" is interesting. Does it indicate that low CRs might be partially due to the unfaithfulness of reasoning models? Have you done any ablation study to compare reasoning vs. non-reasoning models on their CRs?

---

### Official Review · Reviewer_LzS6 · 2025-11-04

**Soundness:** 2
**Presentation:** 2
**Contribution:** 2
**Rating:** 2
**Confidence:** 4

**Summary:**

This work highlights a gap in problem solving versus information seeking capabilities of modern Large Reasoning Models (LRMs). This gaps results from problem solving oriented training of these models which rewards them for arriving at a final answer, even at the expense of making fickle / completely wrong assumptions when provided with insufficient information to solve a question. The authors first create a dataset in order to measure the information seeking abilities of the models. This is done by artificially introducing two types of discrepancies in existing problem-solving evaluation datasets. Next, the authors design appropriate metrics for measuring information seeking capabilities of the models and evaluate a range of frontier models on this task, highlighting the weak performance of models on this task. Finally, the authors curate a dataset for training models to ask for information, via SFT and demonstrate that the finetuned models show improvements on the information seeking task.

**Strengths:**

* I believe that the premise of this work - training of models being heavily geared towards problem solving, overlooking other important aspects such as information seeking / question generation - is important and the work is a good step in addressing that that.
* The paper includes an analysis of results is detailed and elaborate
* Interesting analysis in Figure 7 which potentially shows that the clarification questions decrease with decrease in model confidence.

**Weaknesses:**

While the problem studied in the manuscript is important, I have concerns about the experimental setup

* Majority of the evaluation considered in this paper seems to be reliant on the LLM-as-a-judge setup and I am not convinced about its reliability, given the open ended nature of the tasks (for eg. Evaluation where the response includes a clarification request, i.e. it is more complex than simply assessing the equivalence of two expressions). The authors provide a Human Evaluation confirming in Appendix B.2 but that is only for the task of generating incomplete problems from existing complete ones.
* The above issue is further exacerbated by the fact that the judgement of whether a string involves asking for "clarification / information" is highly subjective. For eg. Considering examples of model responses given in Appendix A, the authors mention that only o3-mini asks for information. However, I would consider all models, except Claude3.7 to have asked for information since all of them explicitly reason about the information being missing and clarify that they are proceeding further under assumptions. I would be interested in knowing the verdict of the LLM-as-a-judge for these specific examples, given that the LLM-as-a-judge prompt in Appendix C doesn't specific anything about the response being terminated after asking for information (which happens only in o3-mini)
* Continuing on the previous two points, I believe that a fair evaluation of information seeking abilities can only be done in the context of the model having access to a tool which lets it query the environment for information. For eg. In the examples given in Appendix A, its likely that models other than o3-mini would have invoked such a tools and waited for response if available. This experiment can be emulated without actually providing such a tool, by prompting the model with a special token that can be output when it needs to query the environment.
* The SFT experiments lack OOD results. These can be obtained by training the models one of the subsets of CRITIC-Math and evaluate it on another.
* The results of SFT experiments need to be validated on other student models to confirm the trends.

### Other comments
* The manuscript contains typos / grammatical errors at several places which hinders readability.
* Line 150-151: Wrong citation for Omni-Math
* Lines 265-266: It should be Tables 3 and 4 instead of 8 and 9

**Questions:**

* The manuscript contains typos / grammatical errors at several places which hinders readability.
* Line 150-151: Wrong citation for Omni-Math
* Lines 265-266: It should be Tables 3 and 4 instead of 8 and 9

---

### Note · Authors · 2025-12-07

I have read and agree with the venue's withdrawal policy on behalf of myself and my co-authors.